



# 1 Soil concentrations and soil-atmosphere exchange of
# 2 alkylamines in a boreal Scots pine forest

**3 A.-J. Kieloaho[1,2], M. Pihlatie[1,2], S. Launiainen[3], M. Kulmala[2], M.-L. Riekkola[4], J.**

**4 Parshintsev[4], I. Mammarella[2], T. Vesala[2,5], J. Heinonsalo[1]**

[1] {University of Helsinki, Department of Food and Environmental Sciences, P.O. Box 56,
FI-00014, Helsinki, Finland}
[2] {University of Helsinki, Department of Physics, Division of Atmospheric Sciences, P.O.
Box 68, FI-00014, Helsinki, Finland}
[3] {Natural Resources Institute Finland, Environmental Impacts of Production, P.O. Box 18,
FI-010301, Vantaa, Finland}
[4] {University of Helsinki, Department of Chemistry, Laboratory of Analytical Chemistry,
P.O. Box 55, FI-00014, Helsinki, Finland}
[5] {University of Helsinki, Department of Forest Sciences, P.O. Box 27, FI-00014, Helsinki,
Finland}
Correspondence to: A.-J. Kieloaho (antti-jussi.kieloaho@helsinki.fi)

## 17 Abstract

Alkylamines are important precursors in secondary aerosol formation in the boreal forest
atmosphere. To better understand the behaviour and sources of two alkylamines,
dimethylamine (DMA) and diethylamine (DEA), we estimated the magnitudes of soil-
atmosphere fluxes of DMA and DEA using a gradient-diffusion approximation based on
measured concentrations in soil solution and in the canopy air space. To compute the amine
fluxes, we first estimated the soil air space concentration from the measured soil solution
amine concentration using soil physical (temperature, soil water content) and chemical (pH)
state variables. Then, we used the resistance analogy to account for gas transport mechanisms
in the soil, in soil boundary layer and in the canopy air space. The resulting flux estimates
revealed that the boreal forest soil with a typical long-term mean pH 5.3 is a possible source
of DMA (170 ±51 nmol m$^{-2}$ d$^{-1}$) and a sink of DEA (-1.2 ±1.2 nmol m$^{-2}$ d$^{-1}$). We also
investigated the potential role of fungi as a reservoir for alkylamines in boreal forest soil. We



found high DMA and DEA concentrations both in fungal hyphae collected from field humus
samples and in fungal pure cultures. The highest DMA and DEA concentrations were found
in fungal strains belonging to decay and ectomycorrhizal fungal groups, indicating that boreal
forest soil, and in particular, fungal biomass may be an important reservoir for these
alkylamines.

## 1  Introduction

Aerosols are important in cooling the atmosphere through increasing the scattering of sunlight
and increasing albedo through cloud formation. In boreal forests, volatile organic compounds
emitted from the biosphere largely drive aerosol formation, and aerosol growth to cloud
condensation nuclei (Kulmala et al., 1998; Kerminen et al., 2010; Riipinen et al., 2012).
Amines have been suggested to be one of the key compounds in the aerosol formation process
(Angelino et al., 2001; Silva et al., 2008; Kurtén et al., 2008; Smith et al., 2009; Yu et al.,
2012; Almeida et al., 2013).
Amines are nitrogenous organic molecules in the form of $NR_3$, where R denotes hydrogen or
alkyl or aryl group. Low-weight alkylamines, which have one to six atom carbon chains
bound to a nitrogen atom, are known to be degradation products of amino-acid-rich
substrates, such as dairy or fish (Ge et al., 2011a). However, the origin of these amine
compounds in natural environments is poorly understood. Sintermann and Neftel (2015)
concluded that flowering of vegetation especially in springtime, and non-flowering vegetation
during growing season are potential sources of alkylamines. Sintermann and Neftel (2015)
suggested that the contribution of fungal sporocarps and decomposing organic matter as
amine sources increases towards the autumn.
Low-weight alkylamines may be produced in soils during the degradation of organic N
compounds, especially amino acid decarboxylation (Yan et al., 1996; Xu et al., 2006). Kim et
al. (2001) and Rappert and Müller (2005) showed that quaternary ammonium compounds
(e.g. carnitine, choline and betaine), often present in soil solution (Warren et al., 2013;
Warren, 2014), could be degraded to alkylamines (trimethylamine, dimethylamine and
monomethylamine) by the soil microbial community using both aerobic and anaerobic
pathways. Sintermann and Neftel (2015) stated that decaying organic matter contains elevated
levels of precursor substances for alkylamine production, hence indicating that decaying
organic matter may be a source of alkylamines.





Concentrations of alkylamines in atmospheric particles and in gas phase are rarely reported
from boreal ecosystems, despite the importance of amines in aerosol formation processes
(Mäkelä et al., 2001; Smith et al., 2009; Kieloaho et al., 2013), mostly due to challenges in
detecting these compounds. Mäkelä et al. (2001) reported elevated concentrations of
dimethylaminium (protonated dimethylamine) during particle formation periods in boreal
forest. In our previous study (Kieloaho et al., 2013), we found the gas-phase alkylamines in
boreal forest air, and we concluded that the seasonal variations in the atmospheric amine
concentrations is linked to vegetation dynamics and soil activity.
Direct flux measurements of alkylamines are difficult to perform and are very rarely made
(Sintermann and Neftel, 2015) due to the high reactivity of amines and lack of suitable
measurement techniques and instrumentation. However, the magnitude of fluxes can be
indirectly estimated if the concentrations of the target compounds in different reservoirs (e.g.
vegetation, soil and atmosphere) are known. In general, the fluxes are driven by a
concentration gradient between the reservoirs, such as ambient air and an aqueous solution.
As follows, gas-phase concentration in soil air can be calculated by assuming equilibrium
between the aqueous solution and the gas-phase above the solution (Farquhar et al., 1980;
Nemitz et al., 2000). Furthermore, the fluxes through a soil-atmosphere boundary can be
estimated using a gradient-diffusion approximation, often presented by an electrical resistance
analogy (Hicks et al., 1987; Seinfield and Pandis, 1998; Sutton et al., 1998).
In this study, we used three layers to estimate the potential exchange of two alkylamines,
dimethylamine (DMA) and diethylamine (DEA), between soil and the atmosphere (Figure 1).
Amine concentrations in boreal forest soil and in fungal hyphae were measured, and used to
estimate potential fluxes of the selected alkylamines from a boreal Scots pine forest soil to the
atmosphre. We hypothesize that by using soil amine concentration data and the resistance
analogy, it is possible to estimate the potential sources and sinks of alkylamines in the soil.
**2   Materials and methods**
**2.1   Study site and supplementary measurements**
Study site is a Scots pine forest at the SMEARII station (Station for Measuring Forest
Ecosystem – Atmosphere Relations) at Hyytiälä (61°84′N, 24°26′E, 180 m a.s.l.) in southern
Finland (Hari and Kulmala, 2005). The forest stand at the SMEARII station is approximately



50 years old and dominated by Scots pine (*Pinus sylvestris* L.) with Norway spruce (*Picea*
*abies* (L.) H. Karst.), birch (*Betula* L. spp.), and European aspen (*Populus tremula* L.), found
occasionally in the understory. The most common plant species at the ground level are
bilberry (*Vaccinium myrtillus* L.), lingonberry (*Vaccinium vitis-idaea* L.), wavy hairgrass
(*Deschampsia flexuosa* (L.) Trin.), and heather (*Calluna vulgaris* (L.) Hull.). The most
common mosses are Schreber's big red stem moss (*Pleurozium schreberi* (Brid.) Mitt.), and a
dicranum moss (*Dicranum* Hedw. sp.) (Ilvesniemi et al., 2009). The soil at the site is Haplic
podzol on glacial till, with an average depth of 0.5-0.7 m.
A half hour average of soil water content (at 0.05 m), soil temperature (at 0.05 m) and above
canopy (at 23 m) friction velocity was used in the calculations of DMA and DEA equilibrium
gas-phase concentrations in soil air, and to calculate DMA and DEA soil-atmosphere
exchange. Soil temperature was measured using PT-100 resistance thermometers, and soil
water content was measured with a time-domain reflectometer (TDR 100; Campbell
Scientific Inc., Logan, UT, USA). A mean pH-value of 5.3 measured over 14-years, and
sampled once per month during snow free period from three replicate suction cup lysimeters
at 2 cm depth in the mineral soil was used. The 10 and 90 percentiles of the soil pH were 4.5
and 6.0, respectively.
The ambient air concentrations of DMA and ethylamine (EA), and DEA were measured at 2
m, below the overstory canopy (Kieloaho et al. 2013) and used in the flux estimation. The
analytical procedure was incapable to resolve DMA and EA, and therefore only the sum of
these compounds is reported, and later referred as DMA concentration. The DMA+EA and
DEA air concentration measurements were conducted from May 2011 to October 2011 by
collecting weekly air samples into phosphoric acid impregnated glass fiber filters described in
detail in Kieloaho et al. (2013). Measured ambient air concentrations of DMA+EA varied
from 0.49 to 6.4 nmol m$^{-3}$, and the mean observed air concentration with standard deviation
was 1.7±1.2 nmol m$^{-3}$ (Kieloaho et al., 2013). The highest concentration of DMA+EA
(6.4±0.83 nmol m$^{-3}$) was measured in October. Ambient air concentration of DEA varied
from 0.02 to 0.63 nmol m$^{-3}$ the mean being 0.26 (±0.22) nmol m$^{-3}$ (Kieloaho et al., 2013).
**2.2   Soil and fungal hyphae samples**
Soil samples were collected at same time in May 2011. The first soil samples were used to
screen the concentrations of amines in the humus layer, mineral soil and visible fungal





hyphae. A 10-liter sample of the humus layer (F/H-horizon) and a 5- liter sample from the
underlying mineral B-horizon were collected. The soil was homogenized and stored at +4°C
(for about day) until three 2 mL samples of mineral soil, humus layer, and visible fungal
rhizomorphic hyphae were collected.
The second soil samples were stored at +4°C until used in the greenhouse experiment where
the effects of soil organic matter decomposing enzymes on nitrogen turnover processes were
studied (Kieloaho et al., 2016). The soil samples were extracted with 1 M KCl, and analyzed
for low molecular weight amines as described in chapter 2.3.
In total 19 different fungal strains, representing 14 different Ascomycete and Basidiomycete
fungal species were grown one by one for six weeks in LN-AS media containing axenic liquid
cultures (Bäck et al. 2010). The strains were divided into four functionally distinct groups:
ectomycorrhiza, ericoid mycorrhiza, endophytes and decay fungi based on their sequence
identification. Individual strains used in this study are listed in Table C1.
Fungal biomass was collected from the liquid cultures using a Miracloth filter, rinsed with
distilled water and stored at -20°C until extracted and analyzed for amines. Agar plugs and
the growth media, used for fungal inoculation in flask cultures, were analyzed separately for
amine concentrations as negative controls.
**2.3   Low molecular weight amine analysis**
Fungal biomass samples and the first set of soil samples were extracted by dynamic
sonication assisted extraction for 20 minutes with flow rate of 0.5 mL min$^{-1}$ (1% aqueous
acetic acid – acetonitrile, 1:1). Samples inserted in extraction chambers made of polyether
ether ketone (PEEK, 5 cm length, i.d. 7.5 mm) equipped with screw caps. After extraction,
samples were filtered through the 0.45-µm syringe filters.  Extraction solvent was pumped
through the extraction chambers, which were immersed in ultrasonic bath (Branson Sonifier
S-250 A, Branson, Danbury, CT, USA) using Jasco PU-980 HPLC pumps (Jasco Corp.,
Easton, MD, USA).
The samples were statically extracted for 30 minutes. Mineral and humus soil samples, 21.8 g
and 16.2 g of fresh weight, respectively, were extracted by sonication with 40 mL
dichloromethane-methanol (1:1) together with 1 mL 1M HCl for 30 minutes. Also, 700 mg
fresh weight of fungal hyphae samples was weighed and extracted with 10 mL of the
extraction solvent with addition of 100 µL 1M HCl. After the extraction, the fungal and





mineral soil samples were evaporated to 5 mL and humus soil sample to 15 mL, and then
filtered through 0.45 µm acetyl cellulose syringe filters.
Low molecular weight alkylamines in extracts from soil, soil fungal hyphae and fungal pure
cultures were analyzed with the analytical method introduced by Ruiz-Jiminez et al (2012).
Soil extracts and cultured fungal biomass extracts were first dansylated. Since dansylated
amines are relatively unstable, derivatized samples were analyzed immediately or within 24
hours. Acetaminophen was used as an internal standard (the final concentration of the
standard was 1 ng at the detector). The derivatization procedure tends to overestimate amine
concentrations but the estimations of the relative amounts to the internal standard of amines
are presumed to be accurate (Ruiz-Jiminez et al (2012).
Analysis of the samples was performed with an Agilent 1260 Infinity liquid chromatograph
coupled via electrospray ionization to an Agilent 6420 triple quadrupole mass spectrometer
(Agilent Technologies, Santa Clara, CA, USA). The initial mobile phase was a mixture of
50% A (water acidified with 1% acetic acid) and 50% B (acetonitrile). Sample volume of 20
µL was injected and a linear gradient to 100% B in 10 minutes was applied. After 7 minutes
in 100% B, mobile phase was decreased to 50% B in one minute. The column was let to
equilibrate before the next injection for 7 minutes in 50% of B. A Hibar HR column
(Purosphere, RP-18, endcapped, 2 µm, 50 mm x 2.1 mm, Merck, Darmstadt, Germany) was
used and the temperature was kept at 40 °C. Ionization parameters were as follows: drying
gas (nitrogen) temperature 300 °C, gas flow 7.5 L min$^{-1}$ and nebulizer (nitrogen) 35 psi. MS1
and MS2 heaters were kept at 100 °C. The dynamic multiple reaction monitoring acquisition
method was applied. MassHunter Quantitative Analysis software B.04.00 was used for data
processing.
To identify amines in the samples the following external standards were used:
isopropylaniline, tripropylamine, 2-amino-1-butanol, DL-2-aminobutyric acid and
diethylamine for the first field soil samples and the soil fungal hyphae, and methylamine,
dimethylamine, ethanolamine, diethylamine, dibutylamine, and *sec*-butylamine for the second
soil samples (Kieloaho et al., 2016) and for pure fungal culture strains.
**2.4   Concentrations of DMA and DEA in soil air**
The concentrations of DMA and DEA in soil solution (aq.) are obtained from the
measurements in the greenhouse experiment on boreal forest soil (Kieloaho et al., 2016), and





assumed to be constant during the whole study period. The DMA and DEA concentrations in
soil solution were 92.3 μmol L$^{-1}$ and 0.296 μmol L$^{-1}$, respectively.
The concentrations of non-dissociated DMA and DEA are calculated from the measured soil
solution concentrations based on reversible acid-base reaction
$R_3N\ (aq) + H^+\ (aq) \leftrightarrow R_3NH^+\ (aq),$     (1)
where $R_3N$ is non-dissociated amine molecule and R denotes either methyl or ethyl organic
side group or hydrogen atom. The dissociation reaction reaches a temperature dependent
equilibrium, which is independent of reactant and reaction product concentrations.
A concentration in soil solution is a sum of non-dissociated ($R_3N$) and dissociated ($R_3NH^+$)
forms of amines. In the first step, using equilibrium thermodynamic principles, the fraction
($f_{R3N}$) of total amine concentration present as non-dissociated form can be estimated (Montes
et al., 2009), when the activity of $R_3N$ and $R_3NH^+$ are assumed to be equal. The activity of
protons [H$^+$] in soil solution is based on the measured pH values. Equilibrium dissociation
coefficients (pK$_a$) for DMA and DEA are 10.3 and 10.5, respectively, and $K_a$ is a negative
logarithm of pK$_a$,
$f_{R3N} = \dfrac{[R_3N]}{[R_3N]+[R_3NH^+]} = \dfrac{1}{1+\frac{[H^+]}{K_a}}.$     (2)
In the second step, the non-dissociated DMA and DEA are partitioned between aqueous
phases and soil air,
$R_3N\ (aq) \leftrightarrow R_3N\ (g).$     (3)
According to Henry's law, the solubility of non-dissociated gas in a solution is directly
proportional to the partial pressure of the gas above the solution
$k_H = \dfrac{c_{R3N}}{p_{soil}},$     (4)
where $k_H$ is Henry's law coefficient, $c_{R3N}$ is non-dissociated aqueous phase concentration and
$p_{soil}$ is a partial pressure of alkylamines in soil gas phase. Due to temperature dependence,
acid dissociation ($K_a$) and Henry's law coefficients were corrected for temperature by Van
t'Hoff equation
$k_{(T)} = k_1 e^{\frac{-\delta H°}{R(T_2^{-1}-T_1^{-1})}},$     (5)





where $k_{(T)}$ is the temperature corrected coefficient, $k_1$ is the coefficient to be corrected, $\delta H^o$ is
the enthalpy change in reaction or phase transition, $R$ is the molar gas constant, and $T_1$ and $T_2$
are temperatures in Kelvins. To take into an account the effect of acid dissociation on the
partitioning of DMA or DEA between the aqueous and gas phases, a temperature corrected
acid dissociation coefficient was used to calculate the effective Henry's law coefficients
according to Seinfield and Pandis (2006)
$$k_{H(T,pH)} = k_{H(T)}\left(\frac{1+[H^+]}{K_{a(T)}}\right), \qquad\qquad (6)$$
where $k_{H(T)}$ is the temperature corrected Henry's law coefficient, $[H^+]$ is measured proton
concentration of aqueous phase and $K_{a(T)}$ is the temperature corrected acid dissociation
coefficient.
Henry's law coefficient, the acid dissociation coefficient, the acid dissociation reaction and
phase change energies were retrieved for DMA and DEA from National Institute of Standards
and Technology Chemistry WebBook (Linstrom and Mallard, 2014).

## 2.5   Estimation of soil-air fluxes of DMA and DEA

The soil-air fluxes ($F$, nmol m$^{-2}$ d$^{-1}$) of DMA and DEA were estimated using flux-gradient
relationship (Figure 1) as
$$F = \frac{C_s - C_a}{r_{tot}}, \qquad\qquad (7)$$
where $C_s$ and $C_a$ are concentrations (nmol m$^{-3}$) in the soil air space and in the atmosphere at
2.0 m above the forest floor, respectively, and $r_{tot}$ (s m$^{-1}$) is the total gas transport resistance,
which includes soil resistance ($r_g$), quasi-laminar boundary layer resistance ($r_b$) and
aerodynamic resistance ($r_a$) in series.
In soil, the gas transport is dominated by molecular diffusion though the air-filled part of soil
matrix. The soil resistance ($r_g$, s m$^{-1}$) in the organic soil layer of depth $\Delta z_s$ (here 0.05 m) is
estimated as
$$r_g = \frac{\Delta z_s}{D_p} = \frac{\Delta z_s}{D_o \theta_a{}^b}, \qquad\qquad (8)$$
where the molecular diffusivity in soil $D_p$ is computed from the molecular diffusivity in free
air ($D_o$), using air-filled porosity ($\theta_a$) to account for the reduced cross-sectional area and



increased path length in the soil relative to free air. The parameter $b = 1.1$ as reported for
humus layer in Glinski and Stepnieswski (1985).
The transport through the quasi-laminar boundary layer at the soil surface is described by the
soil boundary-layer resistance ($r_b$, s m$^{-1}$) following Schuepp (1977)
$r_b = \frac{Sc - \ln(\delta_o/z_1)}{k_v u_{*g} z_1},$ (9)
where $Sc$ is the Schmidt number, $k_v$ (~0.41) is the von Kárman constant, $u_{*g}$ is the near-
ground friction velocity, the height above the ground, where the molecular diffusivity and
turbulent transport efficiency equal, is $\delta_o = D_o/k_v u_{*g}$, and $z_1$ is the height below which the
wind profile is assumed logarithmic. The model for $r_b$ applied here is identical to that used to
compute gas-transfer e.g. in Baldocchi (1988), Nemitz et al. (2001) and Launiainen et al.

11   (2013).

The aerodynamic resistance ($r_a$) accounts for the turbulent gas transport between the soil
surface and concentration measurement height ($z_m$) in the canopy air space. The $r_a$ is
calculated by integrating the inverse of eddy diffusivity ($K_s$, m$^2$ s$^{-1}$) over the layer as in
Baldocchi (1988)
$r_a = \int_0^{z_m} \frac{1}{K_s(z)} dz.$ (10)
The profile of $K_s(z)$ within the canopy and the value of $u_{*g}$ needed for computing $r_a$ and $r_b$,
are provided by a first-order closure model for momentum exchange within the canopy as in
Launiainen et al. (2013, 2015). As shown in Supplement B, the model computes mean
velocity, momentum flux $\overline{(u'w')}$ and $K_s$ profiles from local balance of momentum absorption
and canopy drag neglecting the effects of atmospheric stability. The latter have been shown
modest for below-canopy flow statistics at the SMEAR II –site (Launiainen et al., 2007).
For DMA and DEA flux estimates, the measured weekly mean ambient air concentrations and
their standard deviations (Kieloaho et al., 2013) were used. Soil air concentrations and total
resistances were obtained from the calculated half-an-hour values and averaged to weekly
means and their weekly standard deviations. Gaussian error propagation was used to estimate
the error of flux estimate with an assumption that errors of concentration gradient ($C_{gr} =$
$C_s - C_a$) and total resistance ($r_{tot}$) are independent from each other. The error, expressed as





standard deviation of soil flux ($F_{std}$), was calculated from normalized standard deviations of
$C_{gr}$ and $r_{tot}$
$$F_{std} = F \sqrt{\left(\frac{C_{gr,std}}{C_{gr}}\right)^2 + \left(\frac{r_{tot,std}}{r_{tot}}\right)^2}. \qquad (11)$$
## 2.6  Chemical reaction and turbulent transport timescales
Ratio between turbulent transport timescale and chemical reaction timescale (Damköhler
number, DA) is a measure of flux divergence due to chemical reactions occurring in the
ambient air. As DMA and DEA are reactive gases, their respective
$$DA = \frac{\tau_{tr}}{\tau_{ch}} \qquad (12)$$
were calculated to compare their atmospheric lifetimes ($\tau_{ch}$) to characteristic turbulent
timescale $\tau_{tr} = r_a/z_m$ which are associated to transport between the soil and the atmosphere,
in this case the within-canopy measurement height. DMA and DEA mainly react in the
atmosphere with hydroxyl (OH) radicals, and the chemical timescales $\tau_{ch}$ for DMA and DEA
are 3.2 h and 2.6 h, respectively (Héllen et al., 2014). DA smaller than unity indicates that
chemical reactions play a minor role in linking measured flux at a given height to
sinks/sources below the measurement height (Rinne et al., 2012). When DA is smaller than
0.1, the role of chemical reactions is typically neglected in flux estimates (Rinne et al., 2012).
## 2.7  Sensitivity analysis
The sensitivities of the calculated resistances and estimated soil air concentrations and soil
fluxes were assessed by one-at-a-time method by studying the effect of the measured variable
on the calculated variable. In case of soil air concentrations, the studied variables were pH
(from 4.0 to 6.0), temperature (from 0 to 20 °C) and soil solution concentration (from 0 to 100
µmol L$^{-1}$), as these variable have an effect on dissociation and separation between gas and
aqueous phases of DMA and DEA. The measured soil solution concentrations were based on
1 M KCl extractions. The soil solution concentration of DMA was used as the upper limit for
the soil solution concentration range.
The effects of environmental variables on resistances were assessed separately for $r_g$, $r_b$, and
$r_a$. In case of the $r_g$, effect of soil water content (from 0.1 to 0.45 m$^3$ m$^{-3}$) was assessed due to
its effect on soil spore space continuum. In addition, soil temperature (from 0 to 25 °C) and





soil depth (from 0 to 0.15 m) were studied as they affect diffusion and the length of the
diffusion pathway. For $r_b$, the effects of temperature (from 0 to 25 °C) and friction velocity
(from 0.1 to 0.15 m s$^{-1}$) were assessed as they have effects on diffusion and thickness of
quasi-laminar layer, respectively. In case of $r_a$, the effect of friction velocity (from 0.1 to 0.15
m s$^{-1}$) was studied as it determines the effectiveness of turbulent transport.

## 7    3    Results

### 8    3.1    Amine contents in soil, soil-derived fungal hyphae, and pure fungal
9         cultures

Concentrations of DEA in humus soil and in fungal hyphae restricted from the humus were
0.3 µg g$^{-1}$ FW and 2.9 µg g$^{-1}$ FW (Table 1), respectively. Amine concentrations in the mineral
soil were below the detection limit of 0.01 µg g$^{-1}$ FW. DMA was not measured from field
samples, as it was not included in standards used for the first soil samples. The results for
other amine compounds (2-amino-1-butanol and DL-2-aminobutyric acid) analyzed from
field samples are presented in the supplementary material (Table A1).
The highest DMA and DEA concentrations in the fungal pure cultures were measured in the
decay fungi (Table 1). DMA concentrations were much higher than those of DEA throughout
the all functional groups, and concentration of DMA varied from 25 µg g$^{-1}$ FW in endophytic
fungi to 360 µg g$^{-1}$ FW in decay fungi. Three out of four most amine containing fungal strains
belonged to ectomycorrhiza. DEA concentrations in soil fungal hyphae (2.9 µg g$^{-1}$ FW),
ectomychorrhiza (2.5 µg g$^{-1}$ FW) and ericoid mycorrhiza (1.9 µg g$^{-1}$ FW) were in similar
range, while the concentrations in humus and mineral soil were markedly lower (Table 1).
Amine concentrations of DMA and DEA and other measured amines (methylamine,
ethanolamine, *sec*-butylamine, and dibutylamine) of individual strains, as well as the mean
amine concentrations of ecological fungal groups, are shown in supplementary material
(Table C1 and Table C2, respectively).

### 27    3.2    Estimated soil air concentrations

Over the study period, the estimated mean soil air concentrations of DMA and DEA with
standard deviation, at mean soil pH (5.3), were 27±5.1 nmol m$^{-3}$ and 0.032±0.006 nmol m$^{-3}$,
respectively. The effect of soil temperature, soil pH and soil solution concentration on amine





concentrations in soil air are shown in Figure 3. The soil air concentration follows the
seasonal trend in soil temperature (Figure 2). For DMA, the mean soil air concentration was
higher than the measured mean ambient air concentration (1.7 nmol m$^{-3}$) during the study
period. For DEA, the mean soil air concentration was lower than the measured ambient air
concentration (0.26 nmol m$^{-3}$).
Sensitivity of estimated soil air concentration to soil solution concentration was assessed
using a soil solution concentration range from 0 to 100 µmol L$^{-1}$. Soil air concentration
changed linearly in the studied range 29 nmol m$^{-3}$ for DMA and 11 nmol m$^{-3}$ for DEA (Figure
3A).
Soil air concentrations of DMA and DEA are highly sensitive to soil pH. The non-linear
relationship is caused by pH-dependency of dissociation of an alkylamine in soil solution (Eq.
2), and partition of an alkylamine between aqueous solution and gas-phase (Eq. 6).
In the measured range soil air concentration change was 680 nmol m$^{-3}$ for DMA and 0.81 for
DEA (Figure 3C).  Soil air concentrations in pH 4.0 were 0.07 nmol m$^{-3}$ for DMA and less
than 0.01 nmol m$^{-3}$ for DEA. In pH 5.1 soil air concentrations for the both compounds starts
to increase rapidly from 10 nmol m$^{-3}$ for DMA and from 0.01 nmol m$^{-3}$ for DEA to soil air
concentrations in pH 6.0, 680 nmol m$^{-3}$ for DMA and 0.81 nmol m$^{-3}$ for DEA.
Soil temperature had a minor effect on soil air concentrations than pH in assessed ranges. The
concentration change in the temperature range was 24 nmol m$^{-3}$ for DMA and 0.03 nmol m$^{-3}$
for DEA (Figure 3B). Sensitivity of soil air concentration was not assessed for soil water
content because it has an effect only to the transport of DMA and DEA through the soil.
Estimated soil air concentration did not correlate with measured ambient air concentration in
case of DMA (r=0.09, p=0.68), but it correlated in case of DEA (r=0.67, p<0.01) (Figure 7A
and 7B, respectively).
**3.3   Resistances and chemical reaction timescale**
The mean total resistance for soil-air pathway (r$_{tot}$) was 13 500 (±2300) s m$^{-1}$ for DMA and 18
500 (±3200) s m$^{-1}$ for DEA The r$_{tot}$ was dominated, i.e. the transfer of the studied amines
mostly limited, by slow diffusion of through the soil matrix (soil resistance, $r_g$). The mean soil
resistance of both gases was ~14 000 s m$^{-1}$ (Figure 4B) hence being 1 and 2 orders of





magnitude larger than quasi-laminar resistance ($r_b$, 1200 s m$^{-1}$) and aerodynamic resistance
($r_a$, 110 s m$^{-1}$), respectively (Figure 4C).
Sensitivity of each resistance component to environmental variables (soil water content,
temperature and friction velocities and in case of $r_g$ organic soil depth) was assessed
separately (Figure D1). In short, $r_g$ increases linearly with length of the diffusion pathway
($\Delta z_s$) and non-linearly with increasing soil water content (eq. 8). The temperature sensitivities
of $r_g$ and $r_b$ are weak in the studied temperature range, and caused by weak decrease of
molecular diffusivity with temperature. The $r_b$ (eq. 9) and $r_a$ both decrease nearly order of
magnitude when the above-canopy friction velocity increases from 0.1 to 1.5 m s$^{-1}$, while the
$r_b$ to $r_a$ -ratio is quasi-conserved. Most of the non-linear decrease of $r_b$ and $r_a$ occurs at $u_*$
below 0.5 m s$^{-1}$ (Figure D1).
For DMA, Damköhler number (DA) ranged from 0.013 to 0.026 and having a mean of 0.019
(±0.004). For DEA, DA ranged from 0.017 to 0.033 with a mean of 0.023 (±0.005). Due to
DA numbers lower than 0.1 the removal of DMA and DEA by chemical reactions in the
canopy air space can be considered negligible for the flux estimates.
**3.4   Estimated soil fluxes**
The mean soil-atmosphere fluxes of DMA and DEA over May to November 2011
measurement period were 170 (±51) nmol m$^{-2}$ d$^{-1}$ and -1.2 (±1.2) nmol m$^{-2}$ d$^{-1}$, respectively
(Table 2). The DMA flux increased from the spring to summer, and then decreased in the
autumn. Unlike in the ambient air concentrations (Figure 2B), there was no autumnal peak in
the estimated DMA fluxes (Figure 5). The seasonal pattern in DEA flux did not follow the
changes in soil temperature or moisture, and the fluxes were negative most of the
measurement period. Several strong and distinct DEA uptake periods were estimated in June,
August and October (Figure 5).
Effects of environmental variables (pH, temperature, soil water content, soil depth, and
friction velocity) on estimated soil fluxes are shown in Figure 6. A linear increase in soil
solution concentration would increase flux from soil to the atmosphere linearly. (Figure 6A).
The pH has strong effect in the partitioning of DMA and DEA between aqueous and gas
phases (Figure 3C), and thus also in the flux estimates (Figure 6B). The fluxes computed for
10 and 90 percentiles of measured soil pH (4.5 and 6.0, respectively) were -0.67 (±0.68) nmol

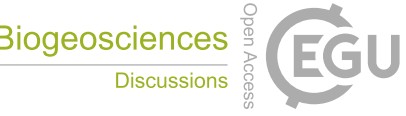



m$^{-2}$ d$^{-1}$ and 4500 (±1300) nmol m$^{-2}$ d$^{-1}$ for DMA, and -1.4 (±1.2) nmol m$^{-2}$ d$^{-1}$ and 2.7 (±1.0)
nmol m$^{-2}$ d$^{-1}$ for DEA, respectively (Table 2).
According to the sensitivity analysis, both amines reach a zero flux point below which the
emission from the soil will turn into an uptake to the soil in the measured pH range from 4.5
to 6.0. This turning point (compensation point with respect to pH) occurred at pH 5.7 for
DEA and at pH 4.7 for DMA was (Figure 6B). A 10% decrease in soil solution concentration
of DMA increased the turning point pH by 0.1 and similarly an increase in soil solution
concentration of DMA decreased the turning point by 0.1 pH unit. The turning point of DEA
was less affected by the soil solution concentration. A change of 10% in DEA soil solution
concentrations lead to a change in turning point pH of ±0.06. Decrease in pH decreased the
available DMA and DEA concentrations and affected partitioning between soil water and soil
air, but the proton concentration had no influence on the transport processes.
Soil temperature increase from 0 to 20 °C increased DMA fluxes from 81 nmol m$^{-2}$ d$^{-1}$ to 255
nmol m$^{-2}$ d$^{-1}$ near-linear manner, and DEA fluxes from -1.1 nmol m$^{-2}$ d$^{-1}$ to 1.3 nmol m$^{-2}$ d$^{-1}$
(Figure 6C) near-linearly. Fluxes decrease near-linearly with increasing soil water content
(Figure 6D). This is due to non-linear increase of $r_g$ with increasing soil water content (Figure
D1). In assessed soil water content range DMA flux changed from 241 nmol m$^{-2}$ d$^{-1}$ to 122
nmol m$^{-2}$ d$^{-1}$ and DEA flux from -1.7 nmol m$^{-2}$ d$^{-1}$ to -0.84 nmol m$^{-2}$ d$^{-1}$ (Figure 6D).
The estimated soil-atmosphere fluxes are sensitive to the assumed depth of amine
sources/sinks in the soil. Because of the dominating role of soil resistance, the absolute value
of flux decrease with soil depth, and the sensitivity is strongest when soil depth is under 0.03
m (Figure 6C) Increasing friction velocity decreases soil boundary layer and aerodynamic
resistances and modestly affect the flux estimates (Figure 6F). The strongest impact occurs
friction velocity values smaller than 0.2 m s$^{-1}$, and is mostly due to $r_b$ (Figure D1). It should
be noted that the friction velocity may become an important factor affecting the flux estimates
in calm conditions if the amine sources or sinks are located very close to the surface leading
$r_g$ and $r_b$ being of same order of magnitude.

## 29    4    Discussion

The results of this study shows that soil is an important reservoir of alkylamines, and our
results suggest that this may be due to high amine concentrations in fungal hyphae in the





boreal forest soil. Furthermore, we show in the flux estimation that these compounds can be
released from the soil into the atmosphere under favorable environmental conditions. The
source-sink behavior was dependent on soil conditions including temperature, soil water
content and pH. Soil was shown to act as a source of DMA and a sink of DEA. The fact that
both the DMA and DEA concentrations were much higher in the fungal hyphae and in fungal
pure cultures as compared to the humus or mineral soil, indicate that the fungal community
may be the primary source of these alkylamines in boreal forest soils.
Both the concentrations of DMA and DEA in humus samples from the greenhouse
experiment (Kieloaho et al., 2016) were lower than those of the fungal pure cultures (Table
1). The DMA concentrations were higher than DEA concentrations in the humus samples and
in pure fungal cultures. Overall, the DEA concentration in the humus samples of the
greenhouse experiment were lower than those measured from the field humus samples (Table

13 1).

In both sample types, field collected hyphae and pure fungal cultures, DEA were found in the
same range strongly supporting each other, and show that fungi are a reservoir of DEA. DEA
concentrations found in the humus soil may reflect concentrations found in fungal biomass
and may be of fungal origin. In the pure fungal culture biomass, DMA concentrations were 50
times higher than those measured for DEA. DMA concentrations were also higher than DEA
concentrations in the soil used in the greenhouse experiment.
Fungal sporocarps were shown to contain of monomethylamine, dimethylamine and
trimethylamine (Sintermann and Neftel, 2015). However, these measurements were based on
fungal sporocarps and not on fungal hyphae, which is the only one form of fungi present in
forest soils. Fungal sporocarps occur seasonally and sporadically mainly in autumn, whereas
fungal hyphae are found throughout the year in forest soil (Santalahti et al., 2016). Therefore,
the sporocarp data does not necessarily reflect the most important fungal contribution as a
source of alkylamines in boreal forest ecosystems.
The fungal community of boreal forest soil undergoes seasonal variation. Santalahti et al.
(2016) observed a clear soil fungal community shift in which the ectomycorrhizal fungi seem
to disappear in late autumn while saprotrophic community dominates in the winter. In this
study we show that ectomycorrhizal fungi contain high quantities of DMA and DEA, which
could be released into the soil solution, and subsequently to the atmosphere during their
disappearance in late autumn. In boreal Norway spruce forest in Sweden, Wallander et al.



(2001) estimated that humus contains 700-900 kg ha$^{-1}$ ectomycorrhizal hyphae, which is equal
to the amount of fine roots found in humus.
The estimated soil air concentrations correlated positively with the measured ambient air
concentrations of DEA, but not with DMA. Kieloaho et al. (2011) found strong correlation
between ambient air concentration of DEA and ambient air monoterpene concentration, and
suggested that the source of DEA might be in vegetation as has been suggested for
monoterpenes (Hakola et al., 2006). In this study, the estimated soil air concentrations of
DEA were smaller than the measured ambient air concentrations, which suggest that the soil
is not necessarily a source of atmospheric DEA. The soil air concentrations are based on
limited data of soil solution concentrations, and the results serve as the first estimates for both
soil air concentrations and soil fluxes for DMA and DEA. DMA and DEA were assumed to
have similar exchange processes with $NH_3$, having both sink and source behavior between the
soil and the atmosphere
At the end of September and in October, the flux estimates of DMA and DEA did not explain
the elevated atmospheric concentrations of DMA and DEA (Figure 2B). This missing
autumnal peak in the fluxes might be due to a rapid change in soil DMA concentration, which
could not be taken into account in the soil air concentration estimates due to the lack of
continuous soil solution concentration measurements. During the autumn (from September to
October), litterfall provides an input of fresh decomposable material into the soil, which also
has an immediate effect on soil nitrogen concentrations due to the nitrogen rich leachate from
the needle litter (Pihlatie et al., 2007; Starr et al., 2014). It was also recently shown that a
common ectomycorrhizal fungal genus *Piloderma* sp., which also contained the highest
quantities of alkylamines in our study, has a clear seasonal pattern, and it seems to disappear
from the soil in late autumn (Heinonsalo et al. 2015). *Piloderma* sp. was shown to be active in
protease production, protease is an enzyme that facilitates the decomposition of proteins,
possibly due to the protease activity *Piloderma* sp. was also found to be able to obtain N from
organic sources and deliver proteinaceous N to the host plant Scots pine. This involvement of
ectomycorrhizal fungi in soil organic N cycling may make them 'nitrogen hotspots' that
release also alkylamines into soil solution after their death (Heinonsalo et al. 2015).
Flux estimates were found to be sensitive to soil temperature, soil pH and soil water content,
and soil resistance had a major effect on transport, while aerodynamic and quasi-laminar
resistances had only minor effects on the fluxes of DMA and DEA. We found that DMA and





DEA flux estimates were especially sensitive to change in soil pH. Flux estimates were
calculated based on three pH values, mean pH (5.3) and 10 and 90 percentiles (4.5 and 6.0,
respectively). The pH, in which the mean flux estimate is zero, is a compensation point with
respect to soil pH. Below the compensation point pH, direction of alkylamine flux is into the
soil and soil is a sink of alkylamines. The compensation point pH occurred for DMA at pH
4.7, which is lower than the mean measured pH from suction lysimeters, indicating that boreal
forest soil can act as a DMA source at least occasionally. In contrary, for DEA the
compensation point with respect to pH was 5.7, which is close to the 90 percentile (pH 6.0),
indicating that soil is a sink of DEA. The compensation point pH is dependent on soil solution
concentration of the amine. Hence, it is clear that even a slight change in soil pH or
alkylamine concentration in soil solution could determine the capability of boreal forest soil
to act as a source or a sink of alkylamines.
Current understanding of the atmospheric alkylamine sources is mainly from rural areas
where the alkylamine emissions are related to agricultural activities (Schade and Crutzen,
1995; Kuhn et al., 2011). Schade and Crutzen (1995) have suggested using a constant ratio
between trimethylamine (TMA) and $NH_3$ in total agricultural emissions as a proxy for
agricultural alkylamine emissions. TMA emissions were 0.3% from $NH_3$ emissions from
livestock farming and it can be partly explained by the same formation pathway of
alkylamines and $NH_3$ (Kim et al., 2001; Rappert and Müller, 2005). The proxy was, however,
revised by Kuhn et al. (2011), who suggested that TMA emissions are 0.1% from $NH_3$
emissions for both livestock farming and vegetation. Mineral soils have been found to be a
sink for atmospheric $NH_3$ while litter of organic layer may act as a source of $NH_3$ (Neftel et
al., 1998; Schjoerring et al., 1998; Nemitz et al., 2000).
It has been proposed that $NH_4^+$ is adsorbed onto soil particles in mineral soil, and hence is not
available for gas exchange between soil solution and gas phase (Neftel et al., 1998). On the
other hand, peat soil and litter layer have been shown to be periodically sources of
atmospheric $NH_3$ in the laboratory (Schjoerring et al., 1998) and in the field (Nemitz et al.,
2000). Previously Hansen et al. (2013) observed $NH_3$ emissions after a litterfall in a
deciduous forest in Denmark, indicating that changes in nitrogen inputs may influence $NH_3$
dynamics. The ambient air measurements of $NH_3$ in boreal forest air indicate that $NH_3$ may be
emitted from the ecosystem in the summer and in autumn as the concentrations of $NH_3$ in
boreal forest air peak during this period, and remain lower in the spring and in winter months





(Makkonen et al., 2014). To our knowledge, the only measured alkylamine fluxes from
forested areas are TMA fluxes measured above a Douglas fir forest from June to July in
Netherlands (Copeland et al., 2014). The mean TMA flux during this one-month
measurement period was around zero showing occasional uptake and emission from -192 to
192 $\mu$mol m$^{-2}$ d$^{-1}$, which is one order of magnitude higher than the DMA flux estimate (170
nmol m$^{-2}$ d$^{-1}$) in this study.
At the moment, ambient air concentration measurements of alkylamines from remote forested
areas are scarce. Recently, there have been several efforts to measure ambient air amine
concentrations using online ion chromatograph connected with quadruple mass spectrometer
(Hemmilä et al., 2014) and CI-API-ToF (Kulmala et al., 2013; Sipilä et al., 2015). However,
they are so far only the first steps in characterizing the amine concentrations and no
continuous datasets are yet available. Flux estimation presented in this study was based on
ambient air concentration measurements conducted by Kieloaho et al. (2013). More recently,
Sipilä et al. (2015) suggested that measured maximum ambient air concentrations of DMA is
0.06 nmol m$^{-3}$ in spring and early summer (from May to June 2013), but due to problems in
measurement system, and lack of calibration they advised to take these numbers by caution.
This implies that if the forest soil is a reservoir of DMA, the real fluxes may be higher than
those presented in this study if the atmospheric concentrations of DMA are as low as those
presented by Sipilä et al., (2015). On the other hand, Hemmilä et al. (2014) reported
preliminary results of ambient air concentrations of DMA and DEA in summer-time (June-
July) at Hyytiälä Scots pine forest to be 0.4 nmol m$^{-3}$ and 0.08 nmol m$^{-3}$ for DMA and DEA,
respectively. These results from June to July indicate that the ambient air measurements by
Kieloaho et al. (2013) are in the correct range. The week long sampling time of ambient air
DMA+EA and DEA concentrations (Kieloaho et al., 2013) coupled with the mixing of air,
atmospheric sink processes and deposition of alkylamines onto the surfaces affect the
measured concentrations, and diminish the relationship between source and ambient air
concentrations. Hence, the flux estimates for DMA and DEA in this study can be used as the
first attempts to estimate potential soil-atmosphere exchange in forests.
The concentration of ammonium in soil water is expected to change with substrate
availability, environmental conditions, microbial activity, and due to assimilation of nutrients
by either soil microbes or vegetation (Pajuste and Frey, 2003). Assuming that DMA and DEA
share similar formation and consumption processes with ammonium in the soil, as suggested





by Kim et al. (2001) and Rappert and Müller (2005), DMA and DEA concentrations in boreal
forest soil may have two maxima during a year, in early spring and in late autumn (Pajuste
and Frey, 2003). The two maxima are due to the combination of supply and demand of
ammonium from temperature dependent ammonium releasing soil processes (decomposition
and mineralization), and plant and microbial uptake rates. In the spring, the decomposition
produces ammonium while the plant-uptake still remain rather low, whereas towards the late
summer, plant uptake exceeds the mineralization rate leading to minimum concentrations in
the soil. In late autumn, plant uptake decreases faster than the mineralization rate leading to a
slight increase in ammonium concentration in soil (Pajuste and Frey, 2003).

## 5   Conclusion

We have shown that boreal forest soil and fungal hyphaea in the soil contain alkylamines,
which can be released to the atmosphere in favourable conditions. We hypothesize that the
soil-atmosphere exchange of the studied alkyamines (DMA and DEA) can be estimated based
on soil temperature, soil water content and especially soil pH. Soil was shown to be a source
of DMA, and a sink of DEA at typical soil pH (5.3) levels. The flux estimation method
presented here is a first attempt to quantify the sources and sinks of alkylamines and other
similar compounds that are difficult to measure directly in forest ecosystems. In boreal forest
soil, fungal hyphae seem to form a large pool of low molecular weight amines like DMA and
DEA. Therefore, we propose that fungi are the origin of alkylamines in boreal forest soils.
The functional role of boreal forest soil as a source of low molecular weight amines, and their
potential emissions needs to be further investigated in relation to air chemistry and
atmospheric aerosol formation processes. In parallel, more measurements on atmospheric and
soil air amine concentrations are needed to confirm the flux estimates provided in this study.

## Acknowledgements

The authors greatly acknowledge Dr. Tiia Grönholm for the help in finalizing this work. This
work was supported by Academy of Finland Centre of Excellence Programme (project
number 1118615), Academy of Finland Research grants (263858, 259217, and 292699), and
The CRAICC and DEFROST Nordic Centres of Excellences.





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



1  Table 1. The mean dimethylamine and diethylamine concentrations (µg g$^{-1}$ fresh weight) and their standard deviations in the field samples,
2  different fungal functional groups and in the humus soil in a greenhouse experiment.

| | Dimethylamine (DMA) | Diethylamine (DEA) |
|---|---|---|
| Field samples | µg g$^{-1}$ FW | µg g$^{-1}$ FW |
| Soil fungal hyphae | n.m. | 2.9 |
| Humus soil | n.m. | 0.3 |
| Mineral soil | n.m. | <0.01 |
| Pure culture samples | | |
| Ectomycorrhiza | 116 (±34) | 2.5 (±0.9) |
| Ericoid mycorrhiza | 80 (±18) | 1.9 (±0.5) |
| Endophyta | 25 (±12) | 0.49 (±0.23) |
| Decay fungi | 360 (±320) | 6.8 (±5.9) |
| Control agar media | 4.3 | 0.13 |
| Experimental samples[a] | | |
| Humus soil (with plant) | 4.3 (±3.9) | 0.03 (±0.02) |
| Humus soil (without plant) | 6.7 (±2.2) | 0.02 (±0.01) |
| Mean of humus soil | 4.6 (±3.2) | 0.02 (±0.02) |

[a] Kieloaho et al. (2016)

n.m.: not measured



1 Table 2. Mean soil air concentrations and flux estimates for dimethylamine and diethylamine followed by their standard deviations in different

2 soil pH values measured in lysimeter waters at 2 cm depth at SMEAR II station.

| soil pH | Dimethylamine (DMA) | | | Diethylamine (DEA) | | |
|---|---|---|---|---|---|---|
| | 4.5 | 5.3 | 6.0 | 4.5 | 5.3 | 6.0 |
| soil air conc. [nmol m$^{-3}$] | 0.68±0.13 | 27±5.0 | 680±130 | 8.1×10$^{-4}$ | 0.03±0.01 | 0.81±0.15 |
| flux/ measured air conc. [nmol m$^{-2}$ d$^{-1}$] | -0.67±0.68 | 170±51 | 4500±1300 | -0.14±0.12 | -1.2±1.2 | 2.7±1.0 |





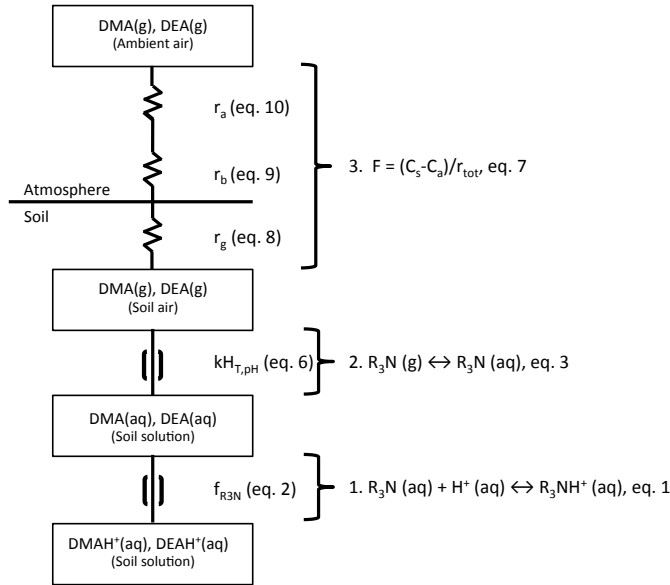

Figure 1. Scheme used for soil-atmosphere flux estimation of dimethylamine (DMA) and diethylamine (DEA) from reactions occurring in soil solution to transfer from soil air to ambient air. Boxes denote DMA and DEA concentrations in soil solution, soil air and ambient air. Numbers denote for steps in the flux estimation. Step 1: acid-base reaction and protonation of alkylamine; step 2: partitioning of non-protonated DMA and DEA between aqueous and gas phases; step 3: flux of DMA and DEA between soil and ambient air in which flux is determined by dividing concentration gradient by sum of component resistances (soil resistance, $r_g$; quasi-laminar boundary-layer resistance, $r_b$; aerodynamic resistance, $r_a$).





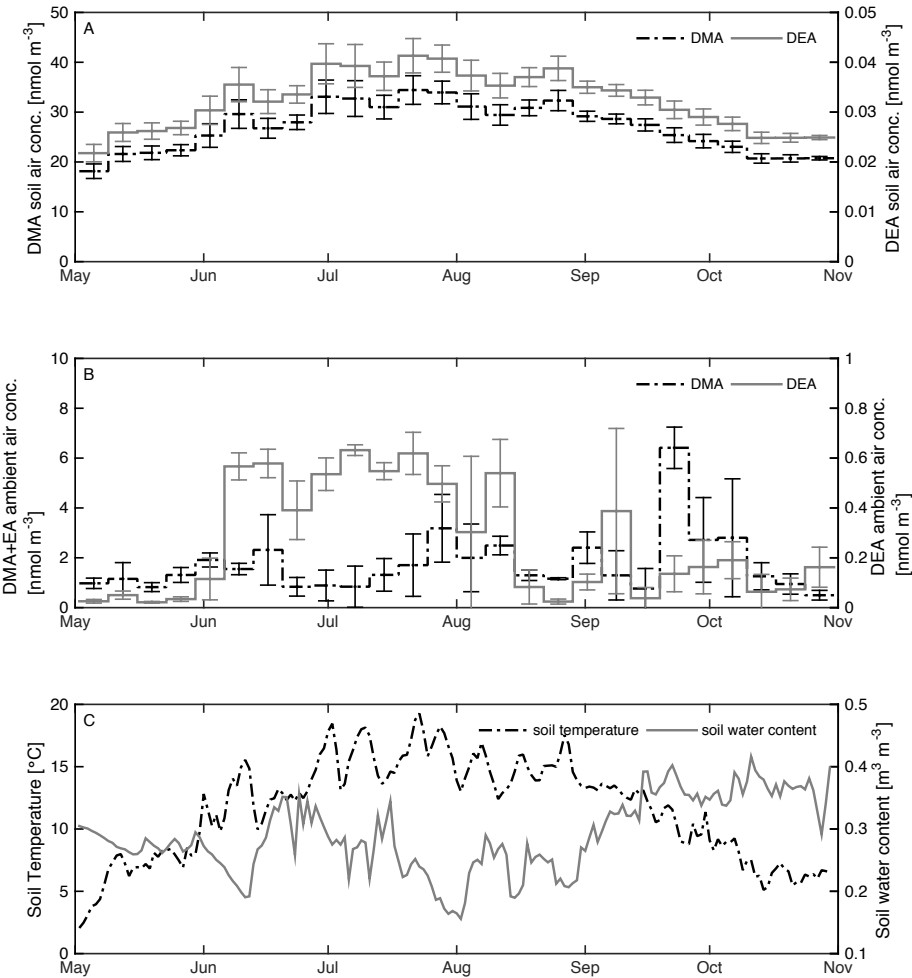

Figure 2. Estimated soil air concentrations of dimethylamine (DMA) and diethylamine (DEA) with their standard deviations at mean soil water pH of 5.3 (panel A), measured ambient air concentration of DMA + ethylamine and DEA (panel B) with their standard deviations redrawn from Kieloaho et al. (2013). In panel C, measured soil temperature and soil water content from May 2011 to October 2011 are shown.





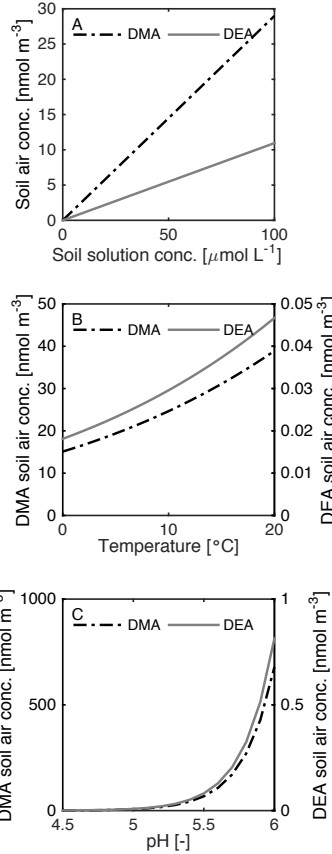

2    Figure 3. Effects of soil solution concentration (panel A), soil temperature (panel B) and soil

3    solution pH (panel C) on estimated soil air concentrations of dimethylamine (DMA) and

4    diethylamine (DEA).





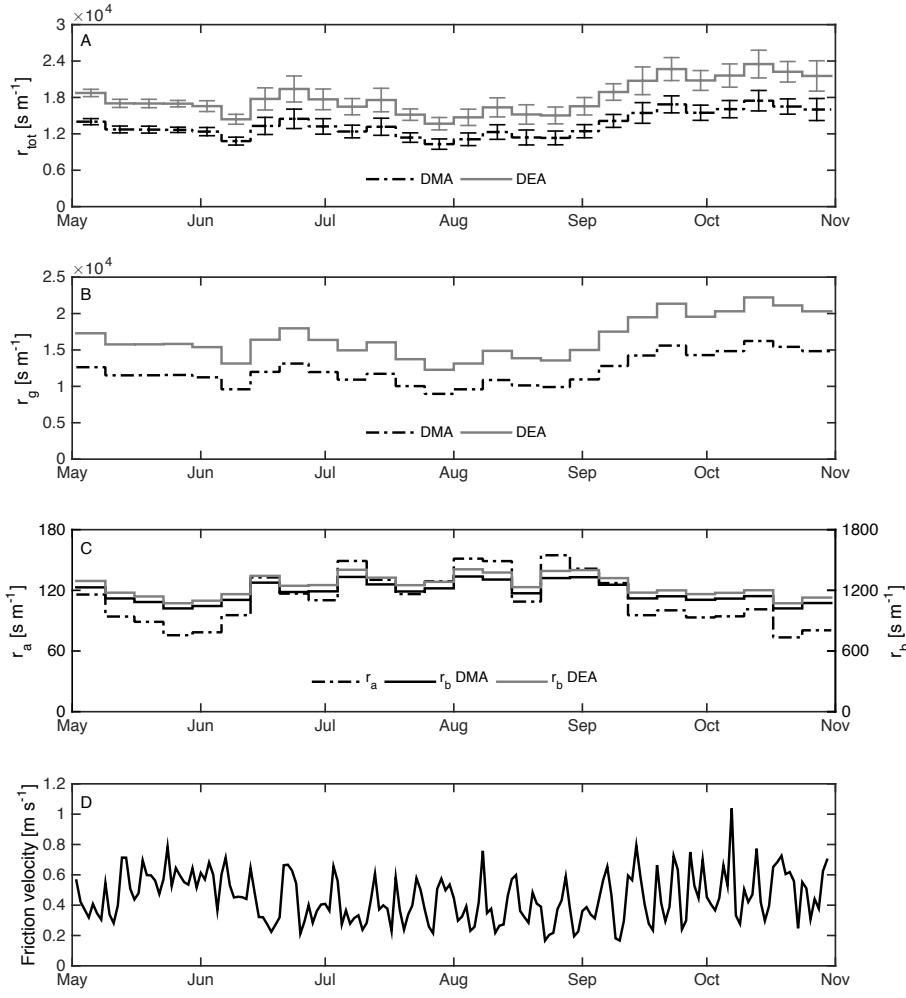

Figure 4. Total resistance ($r_{tot}$, s m$^{-1}$) of dimethylamine (DMA) and diethylamine (DEA) and

its components: soil resistance ($r_g$, panel B) aerodynamic resistance ($r_a$, s m$^{-1}$, panel C), quasi-

laminar resistance ($r_b$, s m$^{-1}$, panel B). Soil resistance ($r_g$, s m$^{-1}$) is calculated using 0.05 m soil

depth. In panel D the daily mean of above canopy friction velocity (m s$^{-1}$) from May 2011 to

October 2011 is shown. The errorbars in panel A show ± one standard deviation.





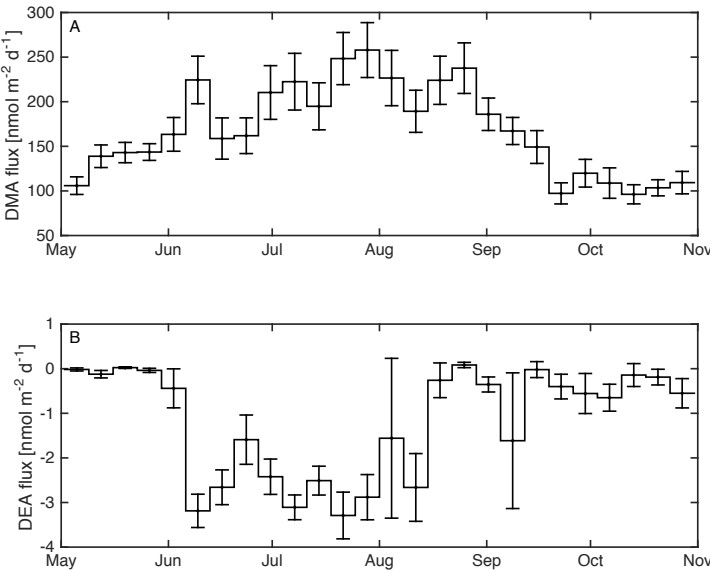

Figure 5. Weekly averages of estimated fluxes of dimethylamine (DMA, panel A) and
diethylamine (DEA, panel B) and their standard deviations from May 2011 to October 2011.
In emission estimation constant soil solution pH 5.3, average soil depth of 0.05 m and
constant soil solution concentrations of DMA and DEA (92.3 μmol L$^{-1}$ and 0.296 μmol L$^{-1}$,
respectively) were used.





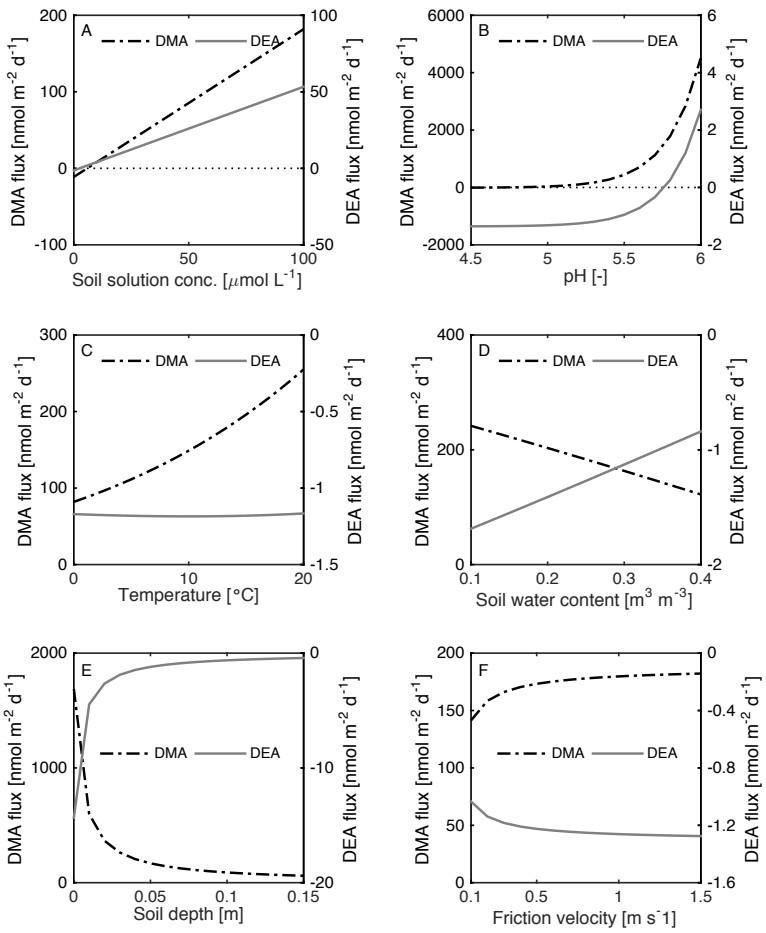

2    Figure 6. Effect of soil solution concentration (panel A), soil pH (panel B), temperature (panel

3    C), soil water content (panel D), organic soil depth (panel E), and above-canopy friction

4    velocity (panel F) on soil flux estimates of dimethylamine (DMA) and diethylamine (DEA).





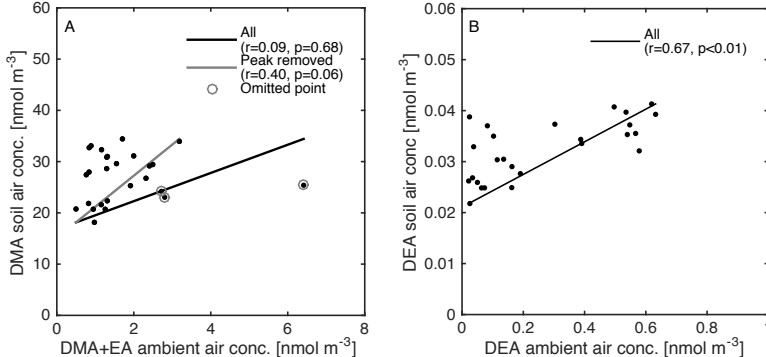

Figure 7. The comparison of measured ambient air concentrations and estimated soil air
concentrations of dimethylamine DMA and diethylamine DEA (panel A and B, respectively)
with linear least square fits. In case of DMA, three data points from autumn have been
omitted (see Sect. 3.3); while also the least square fits without removed points are shown for
comparison.

