# Peer review of "Soil concentrations and soil-atmosphere exchange of"

_Biogeosciences, 2016_

## Referee Comment (RC1) · A. Neftel (Referee) · 20 Sep 2016

The authors discuss the role of boreal forest soil layers as amine source. There is a striking in balance between the apparent importance that amines play in the context of aerosol formation and the knowledge on the emissions. The study focus on fungi as a potential source and presents an estimation of potential exchange fluxes of two amines (DMA and DEA) that have been experimentally accessible. The authors follow a reasonable simple strategy and estimate the fluxes based on a resistance analogy between the concentration in the atmosphere above the soil and the concentration in the open pore space of the soil. The paper is within the scope of BG. An important result is the evidence that fungi in soil are a potential amine source and as fungi are generally part of the organic part of a soil system, soil surfaces can potentially emit amines. Atmospheric concentrations 2m above ground are available with weekly sam-

ples. The soil concentration used in the resistance analogy is calculated assuming equilibrium conditions over a water-air interface with given pH and temperature. The aqueous concentration is determined based on bulk extraction techniques of soil samples and in the laboratory grown fungal samples. I haven't seen from which depth interval the soil samples have been taken. I also cannot judge whether the given values are representative and in the same order of magnitude as what effectively occurs in nature. But the assumption of a single pore space concentration values logically reduces the calculated dynamic of the concentrations in the open pore space over the reported time frame to variability in soil pH, soil water content and soil temperature.

The analysis drastically shows that the depth of the humus layer has the strongest influence on the estimated exchange flux (see figure 6E). This is a consequence of the chosen approach as with the resistance analogy the soil source is assumed to take place at the bottom, i.e. the amine molecules must diffuse through a soil layer with a thickness $\Delta z$ and rg sharply increases with increasing $\Delta z$. I rather think that potential amine sources are distributed in the humus layer proportionally to the decaying rate of fungi. I can also imagine that there are existing consumption processes of amines, so that most of the amines that enter the open pore space will be consumed before they have the chance to reach the atmosphere. The assumed mean layer of 5cm could be a reasonable compromise to yield numerically good looking fluxes.

All in all, I am not fully convinced that the soil in Hyytiälä act as the amine source that drives the measured concentration at 2m in the trunk space. It would be important to directly determine e.g. DMA concentration at the soil surface to give evidence for an emission gradient. The new generation of "ptr-qitof" systems promises to have sensitivities below 1 ppt that should be sufficient to detect a gradient. But of course this is a recommendation for future work and I am also aware tat this systems are very expensive.

A last point: I converted the mean DMA flux of 170nm m-2 and d-1 to roughly 9 gr ha-1yr-1 as I am more used to judge N fluxes per hectare. It would be helpful if this

number is discussed in the context of the yearly N turnover in Hyytiälä. I assume that the vegetation at this station is generally N limited and that the biological systems are using N economically. If I assume the typical ratio of /NH3 of 1% that is found in agricultural systems, total reduced N emissions of the soil compartment would be around 1 kg ha-1yr-1. Is this plausible?
* * *

---

## Referee Comment (RC2) · Anonymous Referee #2 · 23 Nov 2016

The authors present an interesting analysis of soil and atmospheric concentrations of some simple alkyl amines, and use the resistance analogy to estimate the fluxes in a forest ecosystem. Because amines are considered to play an important role in new particle formation, more information about their sources and sinks is valuable. A novel aspect is the identification of fungal hyphae as an important source of amines in the soil. Overall, I think the paper is a useful contribution and should be published after some minor changes.

The main weakness of the paper is the assumption that the soil solution concentrations of amines are constant over the entire May-Oct period, and representative of the study area. This has a major impact on the quantitative (and possibly qualitative) conclusions and does not seem to have been validated in any way. Is this assumption at least consistent with the magnitude of the emissions estimated for DMA (i.e. are fluxes of

the size likely to deplete the soil pool over the measurement period, in the absence of other processes)?

Another drawback of the analysis is that the time resolution of the atmospheric samples (weekly integration) is much lower than the timescale of variability in the conditions that drive the fluxes. Therefore the authors are forced to assume that the average concentration holds throughout the integration period, which is almost certainly not the case. I think one additional sensitivity study would help in assessing how much uncertainty this introduces to the flux estimates. For example, if an artificial diurnal cycle could be imposed on the atmospheric concentration data (giving the same average concentration), with a factor of two difference in concentrations between noon and midnight, how would this affect the calculated fluxes?

Specific comments It should be clarified in the abstract that the mixing ratio attributed to DMA could also have contributions from EA. Section 2.3 - What procedures were used to confirm that the target amines were stable in the extraction procedures described? Perhaps more relevant, can you be sure that there's no contribution from larger molecules degrading to release these simple amines during the extraction procedure? Section 2.4 - How reasonable is the assumption that the soil solution concentrations are constant over the entire May-Oct period, and representative of the study area? This has a major impact on your conclusions and does not seem to have been validated in any way.

Technical comments L24 – atmosphere is misspelled For the Sipila paper, the reference is to the Discussion rather than final version

---

## Author Comment (AC1) · 6 Jan 2017

"The main weakness of the paper is the assumption that the soil solution concentrations of amines are constant over the entire May-Oct period, and representative of the study area. This has a major impact on the quantitative (and possibly qualitative) conclusions and does not seem to have been validated in any way. Is this assumption at least consistent with the magnitude of the emissions estimated for DMA (i.e. are fluxes of the size likely to deplete the soil pool over the measurement period, in the absence of other processes)?"

You are right; one clear weakness in our study is the assumption that the soil solution concentrations of amines are constant. We had discussion on that issue already when we started to work with this project, and we acknowledge that this assumption simplifies

the true condition. However, as the amine concentration measurements in any media (atmosphere, soil, vegetation, fungi) are very rare or nonexistent, and as our study is the first to present amine concentrations in fungal biomass and in boreal forest soil, we decided to keep the estimation scheme simple and approach straightforward. This decision is based on the lack of knowledge in production and consumption processes of amines in the soil-plant systems – as clearly mentioned in the manuscript.

It should be noted that our study is the first one where amine concentrations in fungal biomass and in boreal forest soils are presented. It is possible that soil solution of amines follows same kind of seasonal pattern as Pajuste and Frey (2003) have suggested for ammonium. In the case of amines, it is known that plants are able to take up at least monomethylamine (Kielland, 1994; Wallender and Read, 1999, and Javelle et al., 1999), however use of amines as a source of nitrogen for plants is not well established (Shiraishi et al., 2002; Vranova et al., 2011). One main result of our study was that we could clearly identify gaps in the knowledge concerning amines exchange between biosphere and the atmosphere and suggest future work to better understand the role of amines in soil-atmosphere exchange. As addressed here, assuming the constant soil concentration is not a weakness but also one of the main results of this study. This issue needs to be studied further in future projects.

What comes to the concerns about depletion of amine pool in soil, the ratio of amines in soil solution vs. in volatile form in ambient air is in our study 100 to 1 for DMA and 1 to 1 for DEA. This means that the pool of DMA in the soil matrix does not change very rapidly due to volatilization, while there seem not to be significant of pool of DEA in the studied soil. In addition, as the fungal hyphae was found a significant pool of amines in our study, based on recent studies on renewal of the fungal hyphae (Pickles et al., 2010; Santalahti et al., 2016), we can be quite confident that the renewal of the fungal hyphal biomass in soil is fast enough to release amines into the soil throughout the growing season. Also if amines are released from soil decomposition processes as suggested by Sintermann and Neftel (2015), we can be confidently assume that

amines are released into the soil throughout the growing season in a rate that outcompetes the loss to the atmosphere. In addition, our data suggests that there seems to be hot periods (e.g. autumn) when even more amines as discussed in this manuscript are released into the soil solution and potentially emitted to the atmosphere. But naturally, this should be validated in future studies, when we have better understanding of soil processes involved in amine exchange, and a longer time series of the soil amine concentrations.

"Another drawback of the analysis is that the time resolution of the atmospheric samples (weekly integration) is much lower than the timescale of variability in the conditions that drive the fluxes. Therefore the authors are forced to assume that the average concentration holds throughout the integration period, which is almost certainly not the case. I think one additional sensitivity study would help in assessing how much uncertainty this introduces to the flux estimates. For example, if an artificial diurnal cycle could be imposed on the atmospheric concentration data (giving the same average concentration), with a factor of two difference in concentrations between noon and midnight, how would this affect the calculated fluxes?"

We did, as suggested, additional sensitivity analysis by introducing artificial sinusoidal diurnal cycle into the weekly ambient air concentrations. As the diurnal cycles for studied amines are not yet fully understood, we introduced two scenarios based on current knowledge. In the first scenario we set ambient air concentration minimum at 4 am assuming that diurnal cycle follows that of air temperature. You et al. (2014) observed temperature dependent diurnal cycle for NH3 and trimethylamine in their measurements in a forest site in Alabama (US). In the second scenario we set minimum at 2 pm assuming amine concentrations behaves as observed for monoterpenes in the studied forest environment by Hakola et al. (2012). In the both scenarios, amplitude of ambient air concentrations was set to be two times the measured ambient air concentrations as suggested.

In the manuscript, the estimated mean DMA flux was 170 ($\pm$51) nmol m-2 d-1 and

DEA flux was -1.2 (±1.2) nmol m-2 d-1 during the study period from May to November. When the artificial diurnal cycles were introduced the DMA flux was 170 (±61.8) nmol m-2 d-1 (Fig. 1 middle) and DEA flux was -1.12 (±2.79) nmol m-2 d-1 (Fig. 2 middle) in the first scenario. In the second scenario the DMA flux was 169 (±55.8) nmol m-2 d-1 (Fig. 1 lower) and for DEA the flux was -1.22 (±2.90) nmol m-2 d-1 (Fig 2. lower) during the study period. In the case of DMA diurnal cycle did not have as great effect on the fluxes estimated in the manuscript. It did however increase the variability as you suspected if minimum is at 4 am. In the case of DEA, diurnal cycle has greater effect on flux estimates. Based on the artificial diurnal cycle it can be that soil can act as a source for DEA. However, at the current knowledge diurnal cycle of the amines is not known and this should be studied further as soon as there is possibility to measure amines more frequently than in weekly concentration measurements conducted by Kieloaho et al. (2013).

Following text was added in the manuscript (P11 L12-L20): The weekly ambient air concentration measurements neglect potential diurnal variation of the studied alkylamines. To assess whether this significantly affects the estimated DMA and DEA fluxes, two different sinusoidal diurnal cycles were introduced. The first scenario assumes the diurnal cycle follows that of air temperature, as suggested for NH3 and trimethylamine in a forest site in Alabama (US) (You et al., 2014). The second scenario assumes that diurnal cycle of alkylamines behaves as observed for monoterpenes at the site of our study (Hakola et al., 2012). Consecuently, the minimum concentrations were assumed to occur at 4 am and 2 pm, respectively, and the amplitude of ambient air concentrations was set to be two times the measured weekly concentration.

Following text was added in the manuscript (P15 L12-L16): The flux estimates were modestly sensitive to assumed diurnal cycle of ambient air concentration. Assuming air temperature –dependent diurnal cycle (scenario 1), the DMA flux was 170 (±61.8) nmol m-2 d-1 and DEA flux was -1.12 (±2.79) nmol m-2 d-1. In the second scenario, which assumes the alkylamines behave as that of monoterpenes, the DMA flux was

169 (±55.8) nmol m-2 d-1 and for DEA the flux was -1.22 (±2.90) nmol m-2 d-1.

Following text was added in the manuscript (P18 L6-L12): The diurnal cycles of ambient air concentrations of the studied amines are still currently unknown. By introducing artificial diurnal cycles as observed for trimethylamine or NH3 (You et al., 2014), and monoterpenes (Hakola et al., 2012), it was found out that the diurnal cycles are not likely to have major effect on estimated DMA flux. However, the unknown diurnal cycle of ambient DEA concentration may significantly contribute of the uncertainty and even to sign of the estimated DEA soil-atmosphere DEA flux.

"It should be clarified in the abstract that the mixing ratio attributed to DMA could also have contributions from EA. "

This is now clarified in the abstract and following sentence was added: Used ambient air concentration of DMA was a sum of DMA and ethylamine.

"Section 2.3 - What procedures were used to confirm that the target amines were stable in the extraction procedures described? Perhaps more relevant, can you be sure that there's no contribution from larger molecules degrading to release these simple amines during the extraction procedure?"

Analytical procedure was validated elsewhere (Ruiz-Jimenez et al., 2012). Recoveries and stability of the analytes were assessed with standard addition method at two concentrations (0.25 and 10 ng per sample). Addition was performed to a pool aerosol sample. According to the results, the analytes were quantitatively recovered and they were stable for the period of the analysis. However, we can never be sure that the studied amines are not produced from other compounds during the sampling, storage or sample preparation, since no relevant/suitable reference material is available. The following clarification was added to the paper (P16 L8-L13):

There is a possibility that degradation of sample compounds results in formation of the studied analytes during the sample preparation procedure. This, however, could

not be assessed, due to the absence of suitable reference materials, thus increasing the measurement uncertainty. Similarly, some of the studied amines could have degraded into smaller compounds and hence not to detected in our analysis, leading to underestimation of the concentrations of the studied compounds. "Section 2.4 - How reasonable is the assumption that the soil solution concentrations are constant over the entire May-Oct period, and representative of the study area? This has a major impact on your conclusions and does not seem to have been validated in any way." As the consumption and release processes of amines in soils are not well established as stated previously, and to keep the estimation method straightforward the effect of different soil solution levels on the fluxes were studied by sensitivity analysis. Based on the results, one of the main reservoirs of amines in the soil is fungal hyphal biomass and as stated in the manuscript fungal biomass is present in large quantity in boreal forest soil (Wallander et al., 1999). In a square meter scale fungal hyphae are present in an almost evenly distributed throughout the forest soil (Pickles et al., 2010) and this biomass is being constantly renewed (Pickles et al., 2010, Santalahti et al., 2016). However, due to significant methodological challenges, very little is known of the fungal hyphal turnover rates in soils. New developments in methodology, based on the use of molecular biological tools and stabile isotopes, and extensive field scale studies are expected to provide more detailed information on fungal hyphal dynamics in boreal forest soils.

To illustrate to complexity of the boreal forest soils, in the Fig. 3 it can be seen how intensively soil is colonized by ectomycorrhizal fungal hyphae. As the turnover rate of this (in the picture mostly white) hyphae may vary from days to months, it is obvious that there are uncertainties related to the assumptions that soil solution concentrations are constant. However, in a stand scale we assume that over any time range, the average amine flux from fungal hyphae to soil may well be rather constant, supporting our assumptions in the manuscript.

"Technical comments L24 – atmosphere is misspelled For the Sipila paper, the reference is to the Discussion rather than final version."

These mistakes are corrected into the text.

"The authors discuss the role of boreal forest soil layers as amine source. There is a striking in balance between the apparent importance that amines play in the context of aerosol formation and the knowledge on the emissions. The study focus on fungi as a potential source and presents an estimation of potential exchange fluxes of two amines (DMA and DEA) that have been experimentally accessible. The authors follow a reasonable simple strategy and estimate the fluxes based on a resistance analogy between the concentration in the atmosphere above the soil and the concentration in the open pore space of the soil. The paper is within the scope of BG. An important result is the evidence that fungi in soil are a potential amine source and as fungi are generally part of the organic part of a soil system, soil surfaces can potentially emit amines. Atmospheric concentrations 2m above ground are available with weekly samples. The soil concentration used in the resistance analogy is calculated assuming equilibrium conditions over a water-air interface with given pH and temperature. The aqueous concentration is determined based on bulk extraction techniques of soil samples and in the laboratory grown fungal samples. I haven't seen from which depth interval the soil samples have been taken. I also cannot judge whether the given values are representative and in the same order of magnitude as what effectively occurs in nature. But the assumption of a single pore space concentration values logically reduces the calculated dynamic of the concentrations in the open pore space over the reported time frame to variability in soil pH, soil water content and soil temperature."

Soil samples were collected from 3 to 5 cm depth in the soil from mixed F and O-horizons. As described in the manuscript, small sample set of field samples were collected. At the time of analysis of field samples only standards for DEA was available. When DEA concentration was compared with the concentrations measured from the experiments, we found out that DEA concentrations were in the same order of magnitude or slightly higher in the field samples than in the samples from experiments.

"The analysis drastically shows that the depth of the humus layer has the strongest influence on the estimated exchange flux (see figure 6E). This is a consequence of the chosen approach as with the resistance analogy the soil source is assumed to take place at the bottom, i.e. the amine molecules must diffuse through a soil layer with a thickness $\Delta z$ and rg sharply increases with increasing $\Delta z$. I rather think that potential amine sources are distributed in the humus layer proportionally to the decaying rate of fungi. I can also imagine that there are existing consumption processes of amines, so that most of the amines that enter the open pore space will be consumed before they have the chance to reach the atmosphere. The assumed mean layer of 5cm could be a reasonable compromise to yield numerically good looking fluxes.

All in all, I am not fully convinced that the soil in Hyytiälä act as the amine source that drives the measured concentration at 2m in the trunk space. It would be important to directly determine e.g. DMA concentration at the soil surface to give evidence for an emission gradient. The new generation of "ptr-qitof" systems promises to have sensitivities below 1 ppt that should be sufficient to detect a gradient. But of course this is a recommendation for future work and I am also aware tat this systems are very expensive."

We agree with You that method we used has drawbacks and it leaves room for discussions. To overcome the restrictions of our straightforward method, we did sensitivity analysis to identify major sources of uncertainties rising from the used estimation method, e.g. we studied effect of depth of amine source in the soil profile. At the present knowledge or according to this study, we cannot conclude that soil processes drive ambient air concentrations of amines. Our approach is the first attempt to identify possible sources in a forest environment. As presented in the manuscript, boreal forest soil contains large and renewing pool of amines in hyphal biomass. According to our results, we can say that it is possible that amines can be released from the soil into the atmosphere. As You stated it is of major importance to study the soil-atmosphere amine exchange further by measuring gradient of amines in different compartments of

boreal forest ecosystems.

Thank You for the tip of the instrument! At the moment, it seems that the measurement techniques are not developed enough to measure gaseous fluxes of amines due to the problems with proton affinity higher than water of these compounds. Measurement techniques utilizing proton transfer reaction (PTR) and hydronium ions as ion source are not suitable for primary or secondary amines. In the case of tertiary amines, proton transfer method using hydronium ions can be used with caution. We would like to thank You for an interesting future topic for studying amines in soil-plant systems. We are aware of a modified version of the PTR technique that uses charged oxygen ions instead of hydronium ions (Sintermann et al., 2011). This technique could potentially be used for amine measurements, but in our knowledge, however, to our understanding it is not commercially available. We are looking forward for more advance techniques utilizing chemical ionization methods and new studies utilizing on-line measurements of amines.

"A last point: I converted the mean DMA flux of 170nm m-2 and d-1 to roughly 9 gr ha-1yr-1 as I am more used to judge N fluxes per hectare. It would be helpful if this number is discussed in the context of the yearly N turnover in Hyytiälä. I assume that the vegetation at this station is generally N limited and that the biological systems are using N economically. If I assume the typical ratio of /NH3 of 1% that is found in agricultural systems, total reduced N emissions of the soil compartment would be around 1 kg ha-1yr-1. Is this plausible?"

If we use suggested 1% for typical ratio of amines and NH3 in agricultural systems, and get the total reduced N emissions of 1 kg ha-1 yr-1, the total reduced N emission is slightly higher than the measured N2O emissions (0.3 kg ha-1 yr-1) from the studied forest soil (Pihlatie et al., 2007; Korhonen et al., 2013). The total reduced N emission value seems to be in reasonable range or at least a good upper estimate as the soil NO3- content at the site is reported negligible while the reduced N (organic and ammonium) content is markedly higher (Korhonen et al., 2013). The highest nitrogen pool

in the studied forest ecosystem is bound to the litter/humus layer (combined F and O horizons; Korhonen et al., 2013), which is approximately 5 to 10 cm thick. In the studied forest site O horizon contained 710 kg N ha-1 and it is approximately 34% of total N pool in the forest (Korhonen et al., 2013). Unlike in agricultural soils, Korhonen et al. (2013) showed that in the studied forest 98.9% of the extractable N is in the form of organic N (26.8 kg N ha-1) and most of the mineral nitrogen is in the form of ammonium (0.31 kg N ha-1).

Based on the N pools in the studied boreal forest environment, we know that the organic N pool is the largest in the whole forest. We also know, based on our earlier studies that mycorrhizal fungi are capable of degrading and utilizing organic N compounds as nutrient source (Talbot and Treseder, 2010). Hence, we hypothesize that soil fungi could also release amines into the soil solution as we demonstrated that they contain high quantities of amines. At the moment the knowledge about the soil solution concentrations of amines (especially in natural systems) are scarce and we cannot say in which ratio amines are present in the soil respect to ammonium or do the amines and ammonium share similar release and consumption processes. Equally likely as assuming a fixed ratio of amine and NH3 emissions, it is possible that fixed ratio with NH3 does not exist. This is topic clearly calls for further studies.

References

Hakola, H., Hellén, H., Hemmilä, M., Rinne, J., and Kulmala, M. (2012). In situ measurements of volatile organic compounds in a boreal forest. Atmos. Chem Phys., 12:11665-11678.

Javelle, A., Chalot, M., Södeström, B., and Botton, B. (1999). Ammonium and methylamine transport by the ectomycorrhizal fungus Paxillus involutus and ectomycorrhizas. FEMS Microbial Ecology, 30:355-366.

Kielland, K. (1994). Amino acid absorption by Arctic plants: implications for plant nutrition and nitrogen cycling. Ecology, 75:2373-2383.

Korhonen, J.F.J., Pihlatie, M., Pumpanen, J., Aaltonen, H., Hari, P., Levula, J., Kieloaho, A.-J., Nikinmaa, E., Vesala, T., and Ilvesniemi, H. (2013). Nitrogen bal- ance of a boreal Scots pine forest. Biogeoscience, 10:1083–1095.

Pajuste, K. and Frey, J. (2003). Nitrogen mineralization in podzol soils under Scots pine and Norway spruce stands. Plant and Soil, 257:237-247.

Pickles, B.J., Genney, D.R., Potts, J.M., Lennon, J.J., Anderson, I.C., and Alexander, I.J. (2010). Spatial and temporal ecology of Scots pine ectomycorrhizas. New Phytologist, 186:755-768.

Pihlatie, M., Pumpanen, J., Rinne, J., Ilvesniemi, H., Simojoki, A., Hari, P., and Vesala, T. (2007). Gas concentration driven fluxes on nitrous oxide and carbon dioxide in boreal forest soil. Tellus Series B – Chemical and Physical Meteorology, 59:458–469.

Ruiz-Jiminez, J., Hautala, S.S., Parshintsev, J., Laitinen, T., Hartonen, K., Petaja, T., Kulmala, M., and Riekkola, M.-L. (2012). Aliphatic and aromatic amines in atmospheric aerosol particles: comparison of three techniques in liquid chromatography-mass spectrometry and method development. Talanta, 97:55-62.

Santalahti, M., Sun, H., Jumpponen, A., Pennanen, T., and Heinonsalo, J. (2016). Vertical and seasonal dynamics of fungal communities in boreal Scots pine forest soil. FEMS Microbiol Ecology, DOI:10.1093/femsec/fiw170.

Shiraishi, T., Kawamoto, Y., Watanabe, T., Fukusaki, E.-I., and Kobayashi, A. (2002). Methylamine treatment changes the allocation of carbohydrate to roots in rice plants. Journal of Bioscience and Bioengineering, 94:460-466.

Sintermann, J., and Neftel, A. (2015). Ideas and perspectives: on the emissions of amines from terrestrial vegetation in the context of new atmospheric particle formation. Biogeoscience, 12:3225-3240.

Sintermann, J., Spirig, C., Jordan, A., Kuhn, U., Ammann, C., and Neftel, A. (2011). Eddy covariance flux measurement technique of ammonia by high temperature chemical ionisation mass spectrometry. Atmospheric Measurement Techniques, 4:599-616.

Talbot, J.M., and Treseder, K.K. (2010). Controls over mycorrhizal uptake of organic nitrogen. Pedobiologia, 53:169-179.

Vranova, V., Rejsek, K., Skene, K.R., and Formanek, P. (2011). Non-protein amino acids: plant, soil and ecosystem interactions. Plant and Soil, 342:31-48.

Wallender, T. and Read, D.J. (1999). Kinetics of amino acid uptake of ectomycorrhizal roots. Plant, Cell and Environment, 22:179-187.

You, Y., Kanawade, V.P., de Gouw, J.A., Guenther, A.B., Madronich, S., Sierra- Hernández, M.R., Lawler, M., Smith, J.N., Smith, J.N., Takahama, S. et al. (2014) Atmospheric amines and ammonia measured with a chemical ionization mass spec- trometer (CIMS). Atmospheric Physics and Chemistry, 14:12181–12194.

[Figure]

**Fig. 1.** Estimated fluxes for DMA. In the upper panel fluxes with standard deviations as presented in the manuscript, in the middle and in the lower panels fluxes with artificial diurnal cycles at minimum 4 am

[Figure]

Fig. 2. Estimated fluxes for DEA. In the upper panel fluxes with standard deviations as presented in the manuscript, in the middle and in the lower panels fluxes with artificial diurnal cycles at minimum 4 am

[Figure]

**Fig. 3.** Illustration of Scots pine rhizosphere and mycorrhizosphere on boreal forest humus.